

# Response of shortwave cloud radiative effect to greenhouse gases and aerosols and its impact on daily maximum temperature

Tao Tang[1], Drew Shindell[1], Yuqiang Zhang[1], Apostolos Voulgarakis[2], Jean-Francois Lamarque[3], Gunnar Myhre[4], Camilla W. Stjern[4], Gregory Faluvegi[5, 6], Bjørn H. Samset[4]

[1]Division of Earth and Ocean Sciences, Duke University, Durham, NC, USA
[2]Department of Physics, Imperial College London, London, UK
[3]National Center for Atmospheric Research, Boulder, CO, USA
[4]CICERO, Center for International Climate and Environment Research, Oslo, Norway
[5]Center for Climate System Research, Columbia University, New York, NY, USA
[6]NASA Goddard Institute for Space Studies, New York, NY, USA

*Correspondence to*: Tao Tang (tao.tang@duke.edu)



**Abstract.** Shortwave cloud radiative effects (SWCRE), defined as the difference of shortwave radiative flux between all-sky and clear-sky conditions, have been reported to play an important role in influencing the Earth's energy budget and temperature extremes. In this study, we employed a set of global climate models to examine the SWCRE responses to $CO_2$, black carbon (BC) aerosols and sulfate aerosols in boreal summer over the Northern Hemisphere. We found that $CO_2$ causes positive SWCRE changes over most of the NH, and BC causes similar positive responses over North America, Europe and East China

but negative SWCRE over India and tropical Africa. When normalized by effective radiative forcing, the SWCRE from BC is roughly 3-5 times larger than that from $CO_2$. SWCRE change is mainly due to cloud cover changes resulting from the changes in relative humidity (RH) and, to a lesser extent, changes in circulation and stability. The SWCRE response to sulfate aerosols, however, is negligible compared to that for $CO_2$ and BC. Using a multilinear regression model, it is found that mean daily maximum temperature (Tmax) increases by 0.15 K and 0.13 K per W m$^{-2}$ increase in local SWCRE under the $CO_2$ and BC

experiment, respectively. When domain-averaged, the SWCRE change contribution to summer mean Tmax changes was 10-30% under $CO_2$ forcing and 30-50% under BC forcing, varying by region, which can have important implications for extreme climatic events and socio-economic activities.

## 1 Introduction

Clouds have a pivotal role in influencing the Earth's energy budget (Ramanathan et al., 1989). By enhancing the planetary

albedo, clouds exert a global mean shortwave cloud radiative effects (SWCRE) of about -50 W m$^{-2}$ at the top-of-the-atmosphere, and by contributing to the greenhouse effect, exert a mean longwave effect (LWCRE) of approximately +30 W m$^{-2}$ (Boucher et al., 2013). On the whole, clouds cause a net forcing of -20 W m$^{-2}$ relative to a cloud-free Earth, which is approximately five times as large as the radiative forcing from a doubling of $CO_2$ concentration. Therefore, a subtle change in cloud properties has the potential to cause significant impacts on climate (Boucher et al., 2013; Zelinka et al., 2017). Recent

studies contended that the cloud feedback, especially the shortwave (SW) cloud feedback, is very likely to be positive (Clement et al., 2009; Dessler, 2010; Zelinka et al., 2017). As the SW cloud feedback is positively correlated with the net climate feedback parameter (Andrews et al., 2012; Andrews et al., 2015; Zhou et al., 2016), a stronger positive SW cloud feedback will lead to higher climate sensitivity and may lead to a future warming towards the high end of current projections (Zhai et al., 2015; Andrews et al., 2018).


On seasonal scales, SWCRE is strongest in the summer months when the solar heating is strongest (Harrison et al., 1990). Because SWCRE is in effect only during daytime, it can substantially modify daily maximum temperature (Tmax). For instance, Dai et al. (1999) found that increased cloud cover can reduce Tmax, thereby decreasing diurnal temperature range. Tang and Leng (2012) reported that the damped Tmax over Eurasia could be partially explained by the cloud cover increase

during 1982-2009. As a positive feedback, SWCRE has also been reported to play a role in heatwave and drought events over Europe by enhancing solar heating (Rowell & Jones, 2006; Vautard et al., 2007; Zampieri et al., 2009; Chiriaco et al., 2014;



Myers et al., 2018). This has influenced the environment, ecosystems and the economy through affecting the frequency and intensity of forest fires, power cuts, transport restrictions, crop failure and loss of life (De Bono et al., 2004; Ciais et al., 2005; Robine et al., 2008). For example, Wetherald and Manabe (1995) reported that in the summer for mid-latitude continents,

higher temperature enhances evaporation in the spring and then evaporation decreases in the summer due to depleted soil moisture. Combined with higher temperature, this summertime evaporation reduction leads to lower relative humidity (RH), which reduces cloud cover and thereby invigorates solar heating. Cheruy et al. (2014) revealed that the inter-model spread of summer temperature projections in Northern mid-latitudes in CMIP5 (Climate Model Inter-comparison Project Phase 5) models is greatly influenced by SWCRE.


All the above studies suggest that the SWCRE plays an important role in influencing the surface energy budget and extreme temperature. Well-mixed greenhouse gases (WMGHGs) and aerosols are currently the two largest anthropogenic forcings (Myhre et al., 2013b). A better understanding on the climate response to these individual forcing agents is increasingly needed, considering their different trends across the globe and opposite impacts on climate (Shindell & Faluvegi, 2009). Due to the

difficulty of separating the forced climate signal of a single agent within observational records, these studies are generally based on model simulations, such as the widely used increased $CO_2$ experiments (Andrews et al., 2012). Many attempts have also been made to explore the aerosol impact on clouds and Earth's energy balance (Lohmann & Feichter, 2005; Chung & Soden, 2017), mean temperature (Ruckstuhl et al., 2008; Philipona et al., 2009), as well as extreme temperature (Sillmann et al., 2013; Xu et al., 2018). However, all these studies treated aerosols as a whole and the individual impacts from absorbing

and scattering aerosols are still less understood. Though some studies investigated the impact from individual aerosol species (Williams et al., 2001; Chuang et al., 2002; Koch & Del Genio, 2010), they generally used only a single model, and the results may be subject to model biases (Flato et al., 2013). Moreover, due to the continuing increase in the likelihood of hot temperature extremes (Seneviratne et al., 2014), as well as their serious consequences (De Bono et al., 2004), it is imperative to have a better understanding on the role of SWCRE from individual forcing agents in hot extremes. However, a multi-model

study on the cloud response to individual aerosol species and the impact of that response on Tmax is still lacking. Given these knowledge gaps, here we investigate the changes of SWCRE to $CO_2$, BC and sulfate aerosols individually and explorex its potential impact on Tmax by using a set of state-of-the-art global climate models. $CO_2$ is the most dominant WMGHG while the latter two represent absorbing and scattering aerosols respectively. This paper will proceed as follows: data and methods are described in Section 2. Results are presented in Section 3, discussions and summary are given in Section 4.

**2 Data and Methods**

**2.1 Data**

This study employs the model output from groups participating in the Precipitation Driver and Response Model Intercomparison Project (PDRMIP), utilizing simulations examining the climate responses to individual climate drivers



(Myhre et al., 2017). The nine models used in this study are CanESM2, GISS-E2R, HadGEM2, HadGEM3, MIROC, CESM-
CAM4, CESM-CAM5, NorESM and IPSL-CM5A. The versions of most models used in the PDRMIP are essentially the same
as their CMIP5 versions. The configurations and basic settings are listed in Table 1. In these simulations, global-scale
perturbations were applied to all the models: a doubling of $CO_2$ concentration ($CO_2 \times 2$), a tenfold increase of present-day black
carbon concentration/emission ($BC \times 10$), and a fivefold increase of present-day $SO_4$ concentration/emission ($SO_4 \times 5$). All
perturbations were abrupt. Each perturbation was run in two parallel configurations, a 15-year fixed sea surface temperature
(fsst) simulation and a 100-year coupled simulation. One model (CESM-CAM4) used a slab ocean setup for the coupled
simulation whereas the others used a full dynamic ocean. $CO_2$ was applied relative to the models' baseline values. For aerosol
perturbations, monthly year 2000 concentrations were derived from the AeroCom Phase II initiative (Myhre et al., 2013a) and
multiplied by the stated factors in concentration-driven models. Some models were unable to perform simulations with
prescribed concentrations. These models multiplied emissions by these factors instead (Table 1). The aerosol loadings in the
CanESM2 model for the two aerosol perturbations are shown in Fig. 1 for illustrative purpose; the spatial patterns are similar
for other models. In the BC experiment, the concentration is highest in East China (E. China), followed by India and tropical
Africa. For the $SO_4$ simulations, the aerosols are mainly restricted to the Northern Hemisphere (NH), with the highest loading
observed in E. China, followed by India and Europe. The eastern US also has moderately high concentrations. More detailed
descriptions of PDRMIP and its initial findings are given in Samset et al. (2016), Myhre et al. (2017), Liu et al. (2018) and
Tang et al. (2018).

**2.2 Methods**

In this study, we focus on the SWCRE at the surface in the low and mid-latitudes during boreal summer months (June-July-
August, JJA hereafter), which is calculated as the difference in the SW radiative flux at the surface between all-sky and clear-
sky conditions (Ramanathan et al., 1989). Changes in SWCRE are obtained by subtracting the control simulations from the
perturbations using the data of the last 20 years in each coupled simulation. The changes are then normalized by the effective
radiative forcing (ERF) in the corresponding experiments to obtain the changes per unit global forcing. The ERF values for
each model are obtained from Tang et al. (2019), which diagnosed those from the data for years 6-15 of the fsst simulations of
each perturbation by calculating the radiative flux changes at the top-of-the-atmosphere (Hansen et al., 2002). The multi-model
mean (MMM) ERF values are $3.65 \pm 0.09$ W m$^{-2}$ ($CO_2 \times 2$), $1.16 \pm 0.25$ W m$^{-2}$ ($BC \times 10$), and $-3.52 \pm 0.63$ W m$^{-2}$ ($SO_4 \times 5$) for
indicated experiments, respectively (MMM$\pm 1$ standard error). Then the MMM changes are estimated by averaging all the nine
models' results, giving the same weighting factor to each model. A two-sided student t-test is used to examine whether the
MMM results are significantly different from zero. The same process was also repeated to other variables analyzed (i.e.,
temperature and humidity).

In order to investigate the impact of circulation changes on specific humidity, following Banacos and Schultz (2005), the
horizontal moisture flux convergence (MFC) is calculated as:



$$MFC = -\nabla \cdot (qV) = -V \cdot \nabla q - q\nabla \cdot V \qquad (1)$$

In Eq (1), q is specific humidity in g kg⁻¹, and V is horizontal wind including both zonal and meridional components. Equation (1) could be further written as:

$$MFC = -u\frac{\partial q}{\partial x} - v\frac{\partial q}{\partial y} - q\left(\frac{\partial u}{\partial x} + \frac{\partial v}{\partial y}\right) \qquad (2)$$

In which $u$ and $v$ are zonal and meridional wind components in m s⁻¹.

## 3 Results

### 3.1 SWCRE Change

Figure 2a-c show the SWCRE changes in response to abrupt changes in $CO_2$, BC and $SO_4$. $CO_2$ causes positive changes in SWCRE over most areas in the NH, indicating that more SW radiation reaches the surface. BC causes similar changes, but
with enhanced (ERF-normalized) magnitude, especially in North America (N. America), Europe and East Asia (E. Asia). In some source regions of BC aerosols (tropical Africa and India), however, the SWCRE changes are negative, which means more SW was reflected. These changes are all statistically significant and are unlikely to be caused by natural variability. Besides, these patterns are also robust and consistent across at least eight of the nine models analyzed under all three experiments (figure not shown). For $SO_4$, the SWCRE changes are relatively small compared with the other two forcings and
few significant changes are found over low-to-mid latitude regions. When domain averaged (green boxes in Fig. 2), the MMM SWCRE from $CO_2$ forcing is, 1.7 W m⁻² (N. America), 2.0 W m⁻² (Europe) and 1.5 W m⁻² (E. China) respectively for the indicated regions. The SWCRE of BC forcing is 7.0 W m⁻² (N. America), 9.0 W m⁻² (Europe) and 9.4 W m⁻² (E. China) respectively, which is roughly 3 to 5 times larger than that from $CO_2$ forcing whereas sulfate aerosols induced 1.2 W m⁻² over E. China and near-zero impact in N. America and Europe, with even the sign of change being uncertain (Fig. 3). Such SWCRE
changes could be largely explained by the changes of cloud cover (Fig. 2d-f). Low-level cloud cover decreased significantly in regions where SWCRE is positive for $CO_2$ and BC forcing, with a stronger decrease from the latter, indicating that the cloud response is more sensitive to BC forcing than to WMGHGs. The sulfate aerosols caused increased cloud cover over mid-latitudes (Fig. 2f). The cloud cover in other levels show similar patterns of change (Fig. S1). In order to better understand these cloud responses, we will explore a set of potential mechanisms driving such changes.


### 3.2 Mechanism of the Cloud Changes

Clouds form when air rises and cools to saturation, and are thus closely linked to changes in RH (Fig. 4a-c). The general pattern of RH changes corresponds well with cloud cover changes (Fig. 2d-f). That is, the cloud cover decreases in regions where the RH drops and vice versa for most areas. A larger RH reduction due to BC compared with $CO_2$ also aligns with a larger cloud cover decrease under BC forcing, especially over N. America and Europe. This spatial pattern is not surprising as it is easier for air masses to reach saturation in conditions with higher RH. By definition, RH depends on both specific humidity and saturation vapor pressure (which, in turn, depends on temperature). To probe which factor determines the RH changes, we further analyzed specific humidity changes (Fig. 4d-f). Specific humidity increases ubiquitously under both $CO_2$ and BC scenarios, as a result of increased evaporation in a warmer climate. Thus, the main driver of the RH drop is the atmospheric temperature that drives a faster increase of saturation vapor pressure. Figure 5 shows the changes of vapor pressure as a function of temperature change over Europe at 850 hPa. For example, the temperature increases by ~1.1 K under $CO_2$ forcing, accompanied by ~0.02 kPa vapor pressure increase. Such a vapor pressure increase, however, cannot keep pace with the rise in saturation vapor pressure, which is about 0.1 kPa. Consequently, the RH decreases in Europe and this is also the case for most other land areas. BC causes stronger temperature increases (and hence larger RH drop) in Europe and N. America, explaining the larger cloud cover reductions compared with $CO_2$. In the source regions of BC, such as India and tropical Africa, the RH increases because of stronger increases of specific humidity, combined with weak or no temperature changes (Fig. S2).

Changes in moisture flux and stability may also play a role in altering specific humidity and cloud formation. Here we analyze the changes of MFC and vertical velocity (omega) and find significant changes under the BC experiment (Fig. 6). It is seen that more moisture is transported to tropical Africa and India (Fig. 6b), which could explain the abovementioned increases of specific humidity in these regions despite their lack of warming. A similar response was noted by Liu et al. (2018), which suggested that more moisture could be brought into monsoon regions due to BC forcing. Koch and Del Genio (2010) noted that BC particles could promote cloud cover in convergent regions as they enhance deep convection and low-level convergence when drawing in moisture from ocean to land regions. This is also observed in our analyses, for example over Africa, North India and Pakistan and part of North China (Fig. 6b and e). However, these impacts may be further compounded by cloud type, circulation, local humidity, and the altitude of BC particles relative to the clouds (Koch & Del Genio, 2010; Samset & Myhre, 2015). The changes in moisture flux and stability in the $CO_2$ experiment are relatively weak compared with those from BC, and most of the changes are only observed in low-latitude regions, possibly due to the shift of Intertropical Convergence Zone (ITCZ) or monsoon circulations. The sulfate aerosols, on the other hand, generally show opposite changes to those from $CO_2$ and BC (Fig. 4c and f), owing to sulfate's cooling effect. The above analyses illustrate that the cloud cover changes could be primarily explained by RH changes and, to a lesser extent, circulation and stability changes.



### 3.3 Fast and Slow Responses

The above responses shown are total responses, which could be further split into fast responses (also called rapid adjustments) and slow responses (Andrews et al., 2010; Boucher et al., 2013). The fast responses generally occur within weeks to a few months with the global mean temperature unchanged with the expectation of a small change over land, which could be obtained by fsst simulations. The slow response is mainly depending on global mean temperature change, which could be estimated by the difference between coupled simulations and fsst simulations, assuming the total response is a linear combination of fast response and slow response (Samset et al., 2016; Stjern et al., 2017). For the $CO_2$ experiment, fast responses dominated in E. US and Europe while both fast and slow responses influence Asia (Fig. 7). When it comes to BC, both fast and slow responses are important in these regions, and in some regions the fast and slow response even show opposite changes (e.g., N. Europe). This is consistent with the findings of Stjern et al. (2017) that the response of cloud amount under BC forcing typically consists of opposite rapid adjustments. Regarding sulfate aerosols, the results are similar to $CO_2$ induced changes, with fast responses dominating in E. US and Europe while both fast and slow responses influence Asia. As discussed in Section 3.2, the slow responses in Asia is likely to be associated with circulation changes, as significant changes in MFC and stability are observed in tropical regions and monsoon regions across all three experiments (Fig. 6). These circulation changes could be, but are not limited to, shifts in the monsoons or ITCZ and tropical expansion, and both greenhouse gases and aerosols have been reported to impact these circulations (Menon et al., 2002; Wang, 2007; Meehl et al., 2008; Seidel et al., 2008; Allen et al., 2012; Turner & Annamalai, 2012).

### 3.4 SWCRE Response to Sulfate Aerosol

Another interesting phenomenon worth noting is the relatively small change in SWCRE induced by sulfate aerosols compared with $CO_2$ and BC. SWCRE at the surface is obtained as the difference of SW fluxes between all-sky and clear-sky conditions (Fig. 8). However, both clouds and aerosol particles scatter solar radiation, so that at least part of the radiation scattered by clouds under all-sky conditions will also be scattered by aerosols under clear-sky conditions (no clouds). This means the SW radiation change at the surface due to scattering may not be as sensitive to cloud fraction changes, which leads to reduced changes in their difference (SWCRE), at least in the source regions (Fig. 8). The SWCRE under sulfate aerosols will not be further discussed due to its small impact.

### 3.5 Impact on Radiation and Tmax

From the energy perspective, the net incoming radiation (Rin) at the surface is the combination of downward SW radiation and downward longwave (LW) radiation minus the reflected SW radiation (Rin = ↓SW − ↑SW + ↓LW). Rin represents the total energy available to maintain the surface temperature and to sustain the turbulent fluxes (Philipona et al., 2009). The surface responds to the imposed Rin by redistributing the altered energy content among the outgoing LW radiation and nonradiative fluxes (ground heat flux and turbulent flux) (Wild et al., 2004). Because SW radiation is in effect only during



daytime while LW radiation works both day and night, Rin is directly related to Tmax. In a perturbed climate, both SW and LW radiation will change, thereby changing Rin and Tmax. The net SW radiation change is further linearly decomposed into SW changes under clear-sky conditions and SWCRE changes. The changes of Rin and its individual components, as well as

Tmax are shown in Fig. 9. For the $CO_2 \times 2$ experiment, the SW under clear-sky conditions shows slight decreases over most of land surfaces, mainly due to the absorption of SW radiation by enhanced water vapor, except for some high-latitude regions where albedo effect is important (Fig. 9a). Combined with the changes of SWCRE and ↓LW radiation, Rin shows significant increases over all land surfaces and thus, increasing Tmax (Fig. 9g and i). The $BC \times 10$ experiment shows similar responses, with significantly negative SW radiation under clear-sky conditions due to SW absorption by BC particles (Fig. 9b) and

enhanced ↓LW radiation resulted from atmospheric heating (Fig. 9f). The resulting Rin changes largely explained Tmax changes on the first order, with cooling observed in source regions (India and tropical Africa) and warming elsewhere (Fig. 9h and j). Nonetheless, some exceptions occurred (i.e., E. China), with decreased Rin but increased Tmax, possibly due to the atmospheric heat transport (Menon et al., 2002) and reduced turbulent fluxes (Wild et al., 2004).

In order to further determine the contributions in Tmax changes from each individual radiative component, a multilinear regression model is applied by regressing Tmax changes to SW clear-sky, SWCRE and ↓LW radiation changes with zero intercept, obtaining the following models:

$CO_2 \times 2$:
Tmax = 0.08 ×$SW_{clear-sky}$ + 0.15×SWCRE + 0.14×↓LW ($R^2$ = 0.73, p < 0.001)

$BC \times 10$:
Tmax = 0.05×$SW_{clear-sky}$ + 0.13×SWCRE + 0.15×↓LW ($R^2$ = 0.80, p < 0.001)

All values in the linear models are MMM changes in each experiment. The models could explain 73% and 80% of the Tmax change in $CO_2 \times 2$ and $BC \times 10$ experiment respectively. The coefficients represent the Tmax change under unit radiative flux change, in which the Tmax increases by 0.15 K (0.13 K) per unit increase in local SWCRE under the $CO_2$ (BC) experiment respectively. A comparison of the original Tmax values and the fitted values from the linear models is shown in Figure 10. The linear models predict the Tmax changes fairly well, with the values scattering along the one-to-one line. The contributions

from each radiative component to Tmax changes were estimated with the linear models and the domain-averaged changes for N. America, Europe, E. China and India (purple boxes in Fig. 9a) are listed in Table 2. Physically, Tmax increases in these regions are mainly due to the increased flux from SWCRE and ↓LW, and partially offset by the reduced flux from $SW_{clear-sky}$ (Table 2 & Fig. 9). Taking N. America under $CO_2 \times 2$ experiment as an example, the warming in Tmax from SWCRE and ↓LW are 0.95 K and 3.24 K respectively, in which SWCRE contributed roughly by 23% to the total warming and the remaining

77% is from the ↓LW radiation change. Such warming is offset by the 0.27 K cooling from SW changes under clear-sky





conditions, leading to a net increase of 3.92 K in Tmax. The contributions of SWCRE in Tmax increases are 29% (Europe), 20% (E. China) and 9% (India) for the indicated regions under the $CO_2 \times 2$ experiment. For the BC×10 experiment, the contributions from SWCRE are larger than those in the $CO_2$ experiment, i.e. 34% (N. America), 47% (Europe) and 34% (E. China) for each region. The response over India under the BC experiment is opposite, in which both SW components cause

cooling in Tmax due to reduced fluxes and such cooling is slightly offset by the warming from increased ↓LW radiation. In this case, the negative SWCRE change contributed 54% to the reduction in Tmax. It is noted that the radiation change might not explain all Tmax changes, as other factors may come into play such as soil moisture, horizontal heat transport and precipitation (Dai et al., 1999). For instance, the temperature response would be different for a dry surface compared to a wet surface, given the same radiative fluxes. This is because more net radiation is realized as sensible heat instead of latent heat

under dry conditions, which has been suggested to play an important role in recent European heatwaves (Seneviratne et al., 2006; Fischer et al., 2007).

## 4 Discussion and Summary

Our study shows that cloud cover in the summer is reduced in a warming climate over most mid-latitude land regions. The reduction of clouds, at the same time, may also reduce the warming effect by reducing downwelling LW radiation (LWCRE,

Fig. S3). Specifically, the LWCRE changes per unit $CO_2$ forcing, in MMM, are -1.1 W m$^{-2}$ (N. America), -0.8 W m$^{-2}$ (Europe) and -1.0 W m$^{-2}$ (E. China) respectively, resulting in net CRE (SWCRE+LWCRE) changes of 0.6 W m$^{-2}$ (N. America), 1.2 W m$^{-2}$ (Europe) and 0.5 W m$^{-2}$ (E. China) at the surface. The LWCRE changes per unit BC forcing are -1.7 W m$^{-2}$ (N. America), -2.1 W m$^{-2}$ (Europe) and -1.5 W m$^{-2}$ (E. China) respectively, leading to net CRE changes of 5.3 W m$^{-2}$ (N. America), 6.9 W m$^{-2}$ (Europe) and 7.9 W m$^{-2}$ (E. China). The net CRE changes are positive under both forcings and work as a positive feedback in these areas. As SWCRE is only active during daytime, the CRE changes have an even more pronounced amplifying effect

on summer extreme temperature in these populated regions.

Recent European heatwave events have been linked to the shift of mean temperature (Schär et al., 2004; Barriopedro et al., 2011). Thus, the enhanced increase in summer mean Tmax may significantly increase the number of hot days and the

probability of heatwave events. Our model simulations show that both N. America and Europe show faster increases in Tmax than in Tmin (daily minimum temperature) under both $CO_2$ and BC experiments (figure not shown), indicating an increase in diurnal temperature range, which has also been reported by Wang and Dillon (2014). These changes can have substantial socio-economic impacts (De Bono et al., 2004; Ciais et al., 2005), influencing human health (Robine et al., 2008), labor productivity (Kjellstrom et al., 2018), and disease transmission (Paaijmans et al., 2010), as well as environmental and other ecological

functions (Vasseur David et al., 2014; Wang & Dillon, 2014).



Some limitations also exist in the current study. Firstly, aerosol-cloud interactions cannot be realistically represented, as more than half of the PDRMIP simulations were run with fixed concentrations, where changes in cloud lifetime cannot affect aerosols. For the BC simulations, three models include aerosol indirect effects (MIROC, NorESM and IPSL) while the remaining ones have only aerosol-radiation interactions included (instantaneous and rapid adjustments). The responses of SWCRE for the two categories are shown in Figure 11. For the regions of interest in the current study, the positive SWCRE over N. America, Europe and E. China and negative SWCRE over India are still observed in the models including indirect effects, but with reduced magnitude. Thus, our main conclusions hold in both sets of models, since the responses do not qualitatively vary between those with indirect effects and models without those effects, though the quantification of the response to BC is model-dependent. Such effects are not likely to be a large source of uncertainty but merit future study. Secondly, the aerosol perturbations are idealized time-invariant $10\times$ and $5\times$ present-day aerosol concentrations. Such simulations provide valuable physical insights into the effects of different forcings on a variety of aspects of the climate system. Aerosol concentrations, however, changed inhomogeneously during the historical period and in recent decades, both spatially and temporally. For example, aerosol concentrations have been decreasing in Europe and N. America since the 1980s and have been increasing in Asia since the 1950s (Smith et al., 2011). Future simulations may use aerosol forcing with realistic spatio-temporal changes.

In conclusion, our study shows that both $CO_2$ and BC could cause positive SWCRE changes over most regions in the NH, with a stronger response caused by BC, except over some key source regions of BC aerosols (e.g., India, tropical Africa) which show opposite changes. The SWCRE changes under sulfate aerosol forcing are, however, relatively small compared with the other two forcers. The SWCRE changes are mainly a consequence of RH changes and, to a lesser extent, circulation and stability changes. The SWCRE changes may have contributed 10~50% of summer mean Tmax increases, depending on forcing agent and region, and contributed substantially to Tmax decreases in the source regions of India and Africa, which has important implications for extreme climatic events and socio-economic activities.

**Acknowledgement**

The PDRMIP model output used in this study are available to public through the Norwegian NORSTORE data storage facility. We acknowledge the NASA High-End Computing Program through the NASA Center for Climate Simulation at Goddard Space Flight Center for computational resources to run the GISS-E2R model and support from NASA GISS. PDRMIP is partly funded through the Norwegian Research Council project NAPEX (project number 229778). A.Voulgarakis is supported by NERC under grant NE/K500872/1. Computing resources for CESM1-CAM5 (ark:/85065/d7wd3xhc) simulations were provided by the Climate Simulation Laboratory at NCAR Computational and Information System Laboratory, sponsored by the National Science Foundation and other agencies.



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



**Table 1.** Descriptions of the nine PDRMIP models used in this study, adapted from Tang et al. (2019).

| Model name | Version | Resolution | Ocean setup | Aerosol setup | references |
|---|---|---|---|---|---|
| CanESM | 2010 | 2.8×2.8 35 levels | Coupled | Emission | Arora et al. (2011) |
| GISS-E2 | E2-R | 2×2.5 40 levels | Coupled | Fixed concentration | Schmidt et al. (2014) |
| HadGEM2-ES | 6.6.3 | 1.875×1.25 38 levels | Coupled | Emissions | Collins et al. (2011) |
| HadGEM3 | GA 4.0 | 1.875×1.25 85 levels | Coupled | Fixed concentration | Bellouin et al. (2011) Walters et al. (2014) |
| MIROC-SPRINTARS | 5.9.0 | T85 40 levels | Coupled | HTAP2 emissions | Takemura et al. (2009) Takemura et al. (2005) Watanabe et al. (2010) |
| CESM-CAM4 | 1.0.3 | 2.5×1.9 26 levels | Slab | Fixed concentration | Neale et al. (2010) Gent et al. (2011) |
| CESM-CAM5 | 1.1.2 | 2.5×1.9 30 levels | Coupled | Emissions | Hurrell et al. (2013) Kay et al. (2015) Otto-Bliesner et al. (2016) |
| NorESM | 1-M | 2.5×1.9 26 levels | Coupled | Fixed concentration | Bentsen et al. (2013) Iversen et al. (2013) Kirkevåg et al. (2013) |
| IPSL-CM | 5A | 3.75×1.9 19 levels | Coupled | Fixed concentration | Dufresne et al. (2013) |

Note: GA = Global Atmosphere. HTAP2 = Hemispheric Transport Air Pollution, Phase 2.







**Table 2.** Domain-averaged Tmax changes from each radiative component estimated from the linear models (unit: K).

| $CO_2 \times 2$ | | | |
|---|---|---|---|
| Region | $SW_{clear-sky}$ | SWCRE | ↓LW | Total |
| N. America | $-0.27 \pm 0.01$ | $0.95 \pm 0.02$ | $3.24 \pm 0.03$ | $3.92 \pm 0.06$ |
| Europe | $-0.24 \pm 0.01$ | $1.14 \pm 0.03$ | $2.79 \pm 0.02$ | $3.69 \pm 0.06$ |
| E. China | $-0.23 \pm 0.01$ | $0.71 \pm 0.02$ | $2.82 \pm 0.02$ | $3.30 \pm 0.05$ |
| India | $-0.29 \pm 0.01$ | $0.26 \pm 0.01$ | $2.59 \pm 0.02$ | $2.56 \pm 0.04$ |
| **BC×10** | | | |
| Region | $SW_{clear-sky}$ | SWCRE | ↓LW | Total |
| N. America | $-0.56 \pm 0.03$ | $1.00 \pm 0.02$ | $1.94 \pm 0.04$ | $2.38 \pm 0.10$ |
| Europe | $-0.73 \pm 0.04$ | $1.15 \pm 0.03$ | $1.32 \pm 0.03$ | $1.74 \pm 0.10$ |
| E. China | $-1.40 \pm 0.08$ | $0.98 \pm 0.02$ | $1.92 \pm 0.04$ | $1.50 \pm 0.15$ |
| India | $-0.89 \pm 0.05$ | $-1.05 \pm 0.02$ | $1.10 \pm 0.02$ | $-0.84 \pm 0.05$ |

Note: uncertainty range was estimated from the 95% confidence interval of each coefficient.






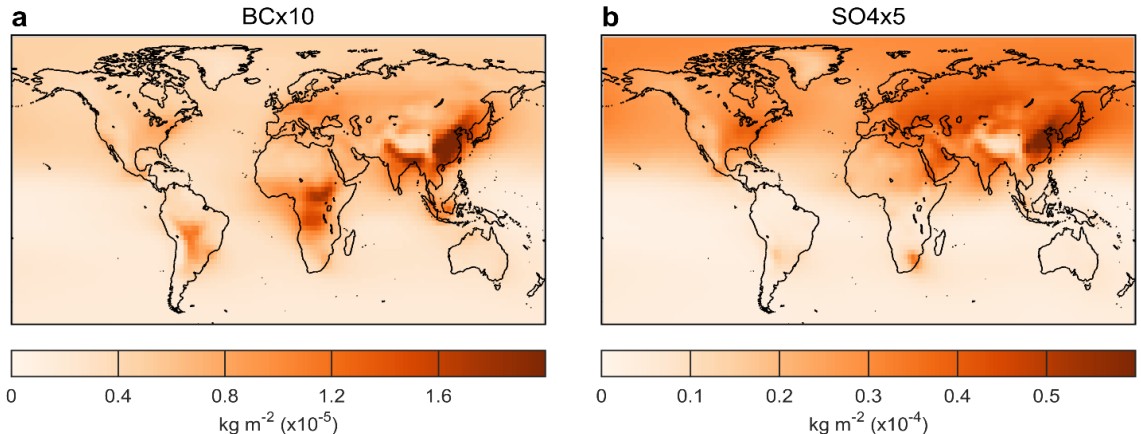

**Figure 1: Aerosol loadings for the two aerosol experiments in CanESM2 model (as an illustrative example).**






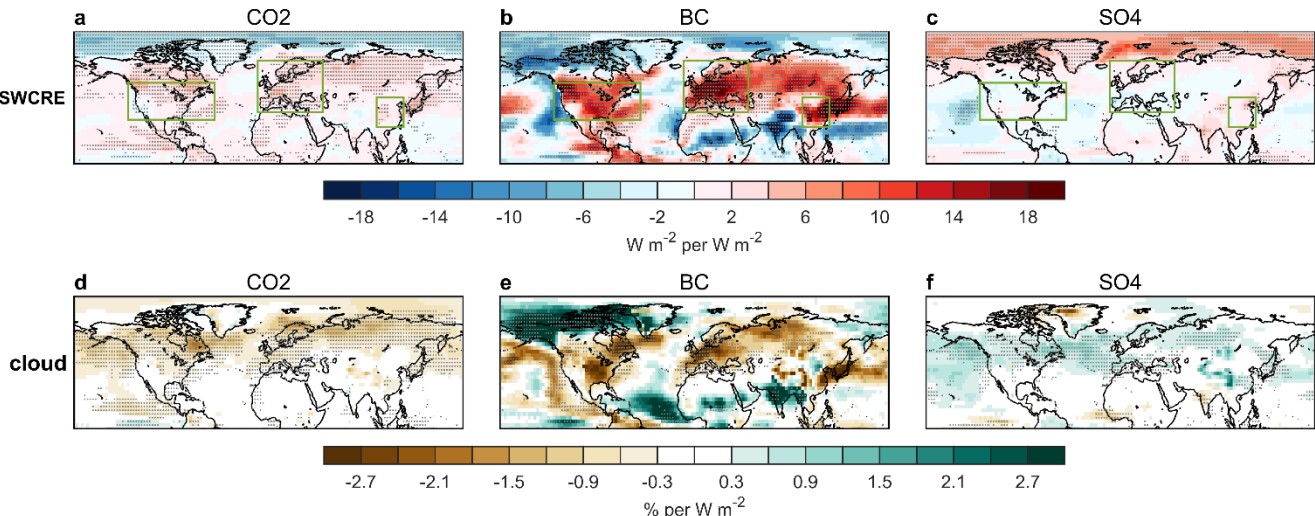

**Figure 2: SWCRE changes (a-c) and cloud cover changes per unit forcing at 850 hPa (d-f) in JJA, results for SO₄ are changes per negative forcing. Grey dots indicate changes are significant at 0.05 level. Positive anomalies in a-c indicate more radiation reaching the surface.**





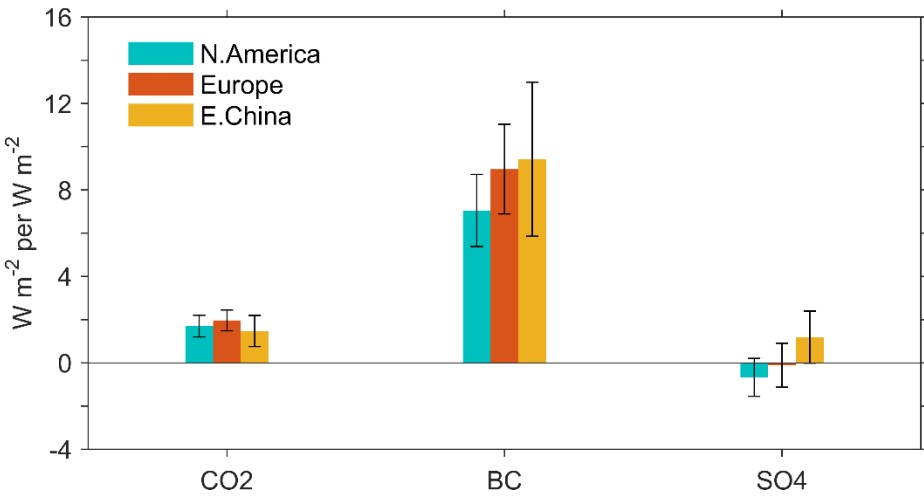

**Figure 3: Domain-averaged SWCRE changes for three regions (green boxes in Fig. 2). Bars represent MMM results and errorbars indicate one standard error across the models.**



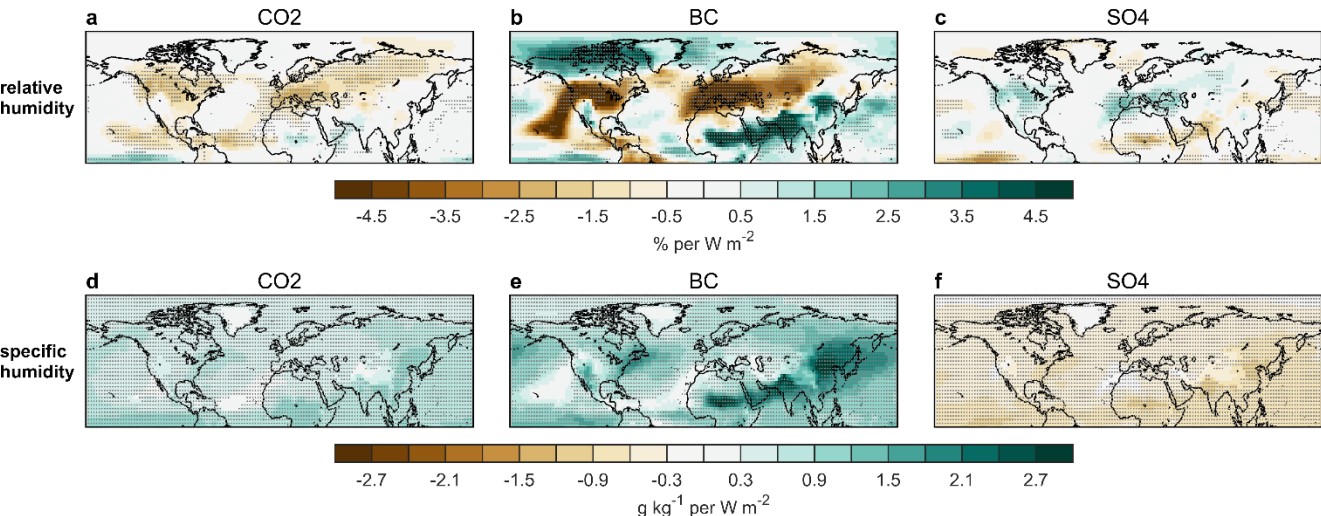

**Figure 4: Same as Figure 2, but for humidity at 850 hPa.**







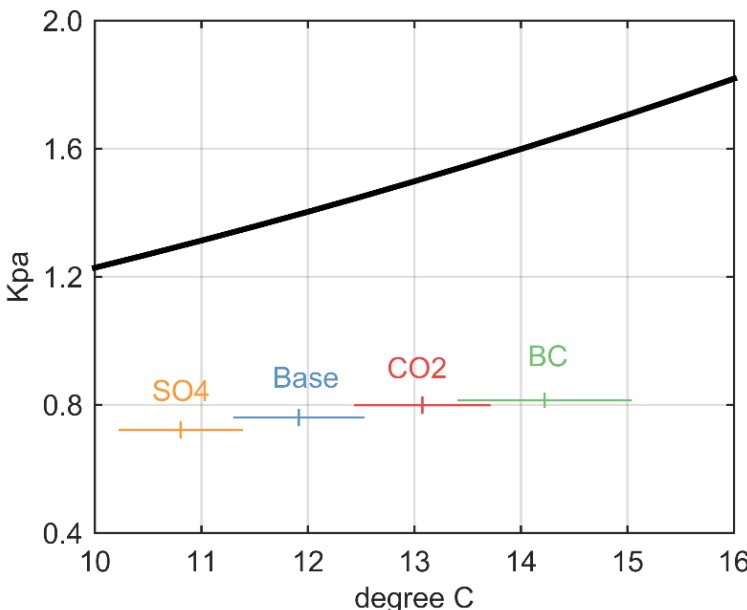

**Figure 5: Domain-averaged vapor pressure changes per unit forcing as a function of temperature at 850 hPa for Europe. Errorbars indicate one standard error across the models. The thick black line represents saturation vapor pressure.**







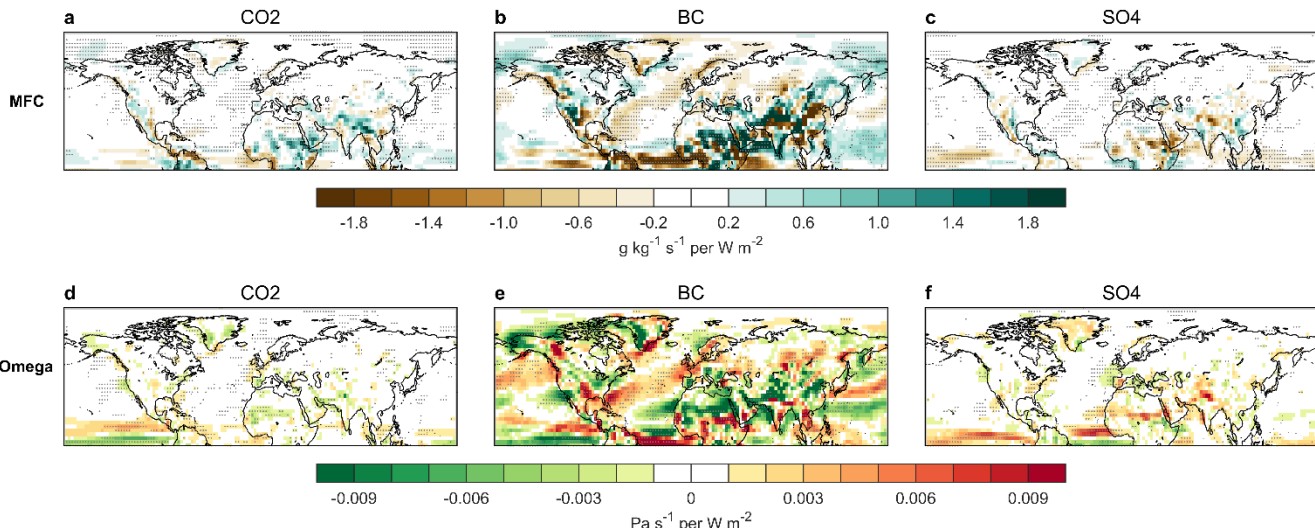

**Figure 6: Same as Fig. 2, but for changes of moisture flux convergence (MFC, a-c) and vertical velocity (omega, d-f) per unit forcing. For vertical velocity, positive anomalies indicate the air is less convective.**







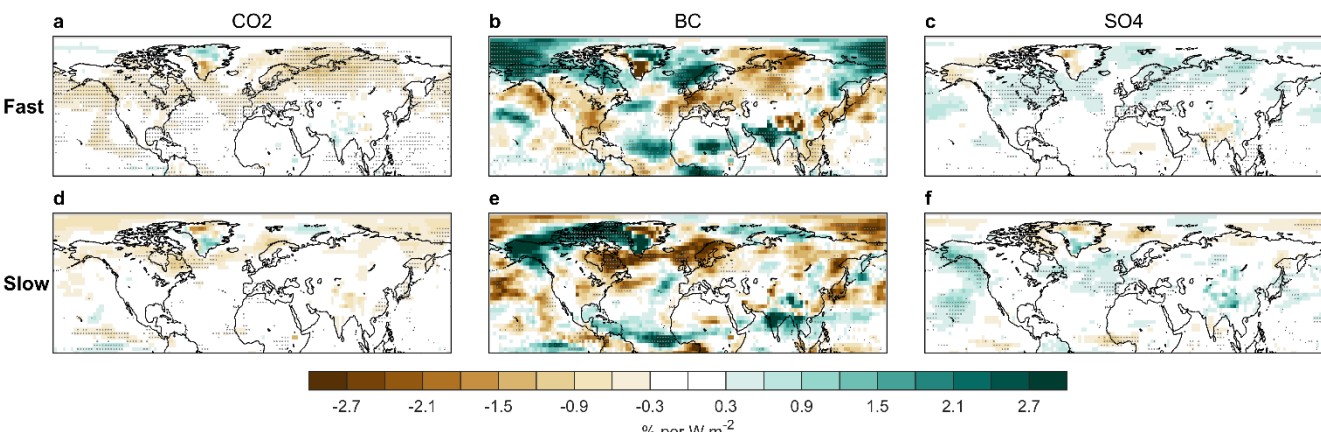

**Figure 7: Same as Figure 2 (d-f), but for fast (a-c) and slow responses (d-f) of cloud cover change per unit forcing.**









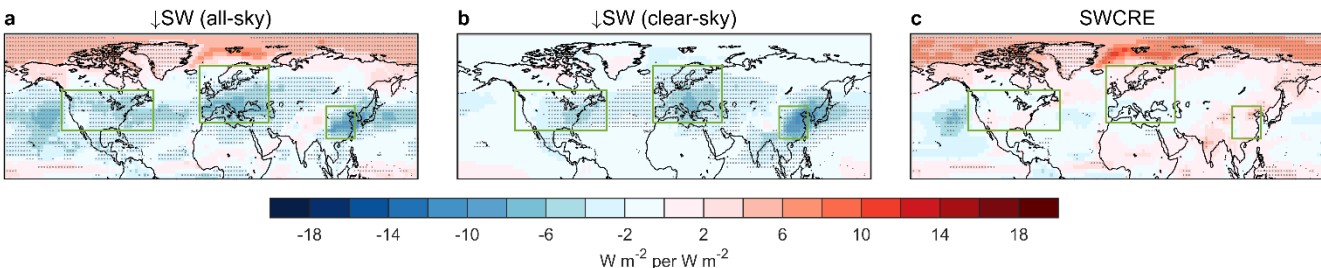

**Figure 8: Changes of SW flux per unit negative forcing under all-sky (a), clear-sky (b) conditions and their difference (c) for the SO$_4$ experiment.**








**Figure 9: Changes of Rin and its components (a-h) as well as changes of Tmax (i-j) for the CO₂×2 (left) and BC×10 (right) experiments (original output, no normalization applied).**



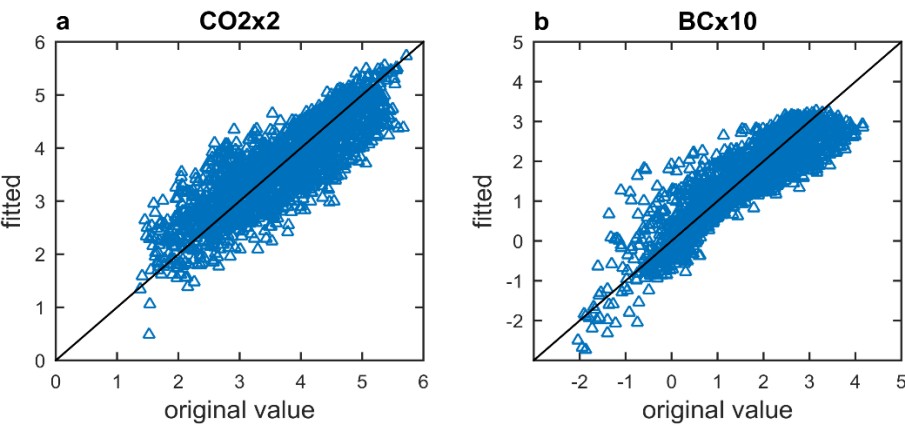


**Figure 10: Comparison of fitted Tmax from the linear models vs original Tmax values. Blue triangles are values for all grid boxes over NH and black solid line represents one-one line.**








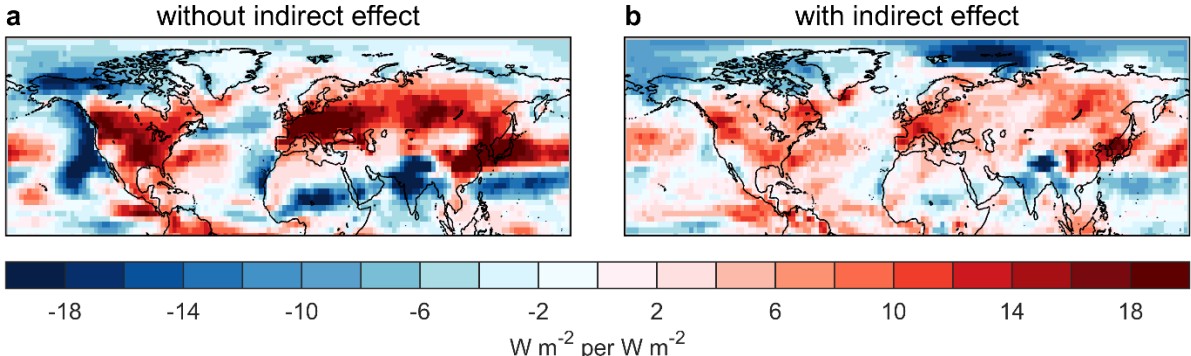


**Figure 11: SWCRE changes for the BC experiment, (a) for models without aerosol indirect effects and (b) for models with indirect effects.**