# Peer review of "Response of shortwave cloud radiative effect to greenhouse gases and aerosols and its impact on daily maximum temperature"

_Atmospheric Chemistry and Physics, 2019_

## Referee Comment (RC1) · Anonymous Referee #1 · 8 May 2020

This paper investigates the response of shortwave cloud radiative effect and daily maximum temperature to greenhouse gases and aerosols (BC and sulfate). It is found that BC results in a stronger positive SWCRE change than CO2 when normalized by effective radiative forcing, but sulfate does not have much effect on SWCRE. It is also shown that the increase in SWCRE resulting from CO2 and BC leads to an increase in daily maximum temperature during the summer. The results are interesting and have some important implications, however a number of things need to be addressed before recommendation for publication.

Major:

1. Most of the results are normalized by effective radiative forcing. What are the surface temperature responses to CO2 and BC, respectively? Could the difference in SWCRE be partly due to the difference in the temperature change (i.e., the efficacy of BC)?

2. The SWCRE change is attributed to the change in cloud cover. I would be interested to see some discussion in the change in cloud liquid water content or liquid water path, which also plays an important role in determining SWCRE.

3. The change in cloud cover is explained by the change in RH. However, there are a lot of other factors affecting clouds (radiation, dynamics, thermodynamics, etc., see Bretherton (2015) and references therein), and I think a more detailed discussion would be helpful. The authors look at vertical velocity and suggest that the change in stability plays less of a role, but it is not clear to me how the conclusion is reached. The estimated inversion strength or lower troposphere stability may be a better predictor for stability.

4. I have some conservation about including downward LW in the multilinear regression model. It is possible that downward LW change is a result rather than a cause of Tmax change (Tmax change results in changes in boundary layer temperature and moisture, and thus downward LW). In fact, consider the approximation LW$\sim\sigma$Tˆ4, dLW$\sim4\sigma$Tˆ3dT, with T$\sim$300 K, dT/dLW$\sim$1/(4$\sigma$Tˆ3)$\sim$0.16, which is very close to the coefficients derived from the regression.

5. In PDRMIP BC and sulfate are increased by a factor of 10 and 5, respectively. It may be helpful to comment on whether the response is linear for such a large change. The small SWCRE response to sulfate is interesting and somewhat surprising. Given that aerosol direct effect is probably more linear than aerosol indirect effect, would the authors expect different SWCRE response to historical change in sulfate?

Minor:

1. Please clarify that the paper analyzes SWCRE at the surface in the abstract, the

main text, and the figures. It is somewhat confusing because I think SWCRE is more commonly referred to as TOA radiative forcing, and the first paragraph in the introduction describes SWCRE at the TOA.

2. Eq.(1): What is the time frequency of q and V for calculating the moisture flux?

3. Figure 7: Maybe show the fast and slow responses of SWCRE instead of cloud cover, as the paper focuses on SWCRE.
* * *

---

## Referee Comment (RC2) · Anonymous Referee #2 · 11 May 2020

The paper is interested in GCM-produced summertime changes in the maximum land temperatures of the NH under perturbed conditions, namely doubled $CO_2$, 10 times more black carbon aerosols and 5 times more sulphate aerosols (subject to model interpretation). Results come from a somewhat outdated database (CMIP5 era models) and the focus is on the SW effects of clouds at the surface (at least initially, later when a prediction model is built LW is added too). I'm not clear what we learn from the analysis. The general consensus since AR5 has been that low clouds provide a positive feedback under $CO_2$ doubling (or quadrupling for that matter), so SWCRE at the surface is expected to be weaker (less cooling at the surface). So, this part is not so new, although I guess one can focus on the effect of this reduced radiative cooling

on Tmax. Then there is the aerosol: aerosol changes can change the environment, the circulation, etc, so they can change cloudiness. But they can impact the clouds "faster" through alteration in microphysics (lifetime, optical thickness changes) and this part is not discussed until the conclusions. In any case, the effect of aerosol on (low?) clouds and therefore on land Tmax is not clear-cut since there is also the direct radiative dimming or brightening part that works in conjunction or competition with the cloud effect. So, it's kind of interesting to see results about this, although I imagine people have previously looked at that too. I guess the most intriguing result is that Tmax changes can largely predicted by LOCAL RADIATIVE changes; it was somewhat unexpected to me that this works as well as it does since temperature is also affected by turbulent fluxes and advection (non-local effects). I suggest the authors make a bigger deal of this finding.

Here are my main issues with this paper: (1) LW and indirect cloud effects are not discussed until the concluding section.

(2) Since the SWCRE effects are mostly attributed to CF changes, the LW downwelling to surface changes should also be broken to clear and LWCRE effects; I mean basically the LW should be treated as the SW and not lumped into a single term in the regression.

(3) Why are only CF changes considered and not changes in other cloud properties? Optical thickness changes can have impact in SWCRE.

(4) Only CF changes for low clouds (and the corresponding RH) are considered (if I understand correctly), but for SW cloud at any altitude in the in the atmospheric column can have strong SWCRE effects

Some minor issues: (1) Clarify from the start (including the abstract) that SWCRE refers to surface.

(2) Define changes in SWCRE more formally. For this SWCRE itself has to be defined more formally, i.e., difference between net all-sky and clear-sky fluxes where net

= down-up flux. Then you have to take a difference between baseline and perturbed conditions. Just saying that a positive SWCRE change means less cooling is unsatisfying.

(3) I find the discussion between fast and slow feedbacks a bit superficial. Land responds to fast feedbacks, but for slow feedbacks the SST responds as well and that's what will drive circulation changes. For slow feedbacks it makes more sense to look at TOA quantities. When it comes to direct radiative effect of aerosol, TOA and SFC changes are distinct for absorbing (BC) vs non-absorbing aerosols (sulphate).

(4) Why not use the same colorbar in Fig. 2 for normalized forcing change and cloud fraction change to make comparison easier (of course range of values can be different)?

(5) By showing only MMM results and nothing about model spread we have no idea how much the models diverge in predictions. Not sure there is an easy way to convey that.

(6) I imagine the radiative treatment of aerosol differs widely among models. Not discussed. When you change emissions instead of concentrations directly, divergence is introduced too.

(7) I also imagine that the base state of the models is quite different too. Care to comment?

---

## Author Comment (AC1) · 16 Jun 2020

**Response to comments #1**

Response: Thanks for your helpful and constructive comments. We have made several modifications and implemented the suggestions as described below. We describe a few major changes first, followed by our response to individual comments.

i) Response of cloud liquid water is added.

ii) Response of lower tropospheric stability is added.

iii) SWCRE response for individual models is added to the supporting material.

iv) Replace Fig. 7 with SWCRE response.

This paper investigates the response of shortwave cloud radiative effect and daily maximum temperature to greenhouse gases and aerosols (BC and sulfate). It is found that BC results in a stronger positive SWCRE change than CO2 when normalized by effective radiative forcing, but sulfate does not have much effect on SWCRE. It is also shown that the increase in SWCRE resulting from CO2 and BC leads to an increase in daily maximum temperature during the summer. The results are interesting and have some important implications, however a number of things need to be addressed before recommendation for publication.

**Major**

1. Most of the results are normalized by effective radiative forcing. What are the surface temperature responses to CO2 and BC, respectively? Could the difference in SWCRE be partly due to the difference in the temperature change (i.e., the efficacy of BC)?
Response: the multi-model mean temperature changes for $CO_2$ and BC experiments are 2.5K and 0.7K respectively. The ratio of 2.5/0.7=3.6 is slightly larger than the ERF ratio (3.65/1.16=3.15), which means if SWCRE changes were normalized by dT, the difference of between $CO_2$ and BC would be slightly larger. As the results would not change much, however, the efficacy of BC will not significantly influence our results in the bar plot (Fig. 3).

2. The SWCRE change is attributed to the change in cloud cover. I would be interested to see some discussion in the change in cloud liquid water content or liquid water path, which also plays an important role in determining SWCRE.

Response: added in Fig. 4 and section 3.2. We added the following discussion after line 172:

[Figure]

**Figure 1: Same as Figure 2, but for humidity at 850 hPa.**

"The response of cloud liquid water in the BC experiment could further support this conclusion (Fig. 4h). Liquid water decreases (increases) in regions with decreasing (increasing) cloud cover, following the pattern of RH. As cloud water content directly impacts cloud optical thickness and albedo, such a response may further impact SWCRE (i.e., enhance reflectance in regions showing increasing liquid water and enhance transmittance in regions with decreasing liquid water). However, the liquid water responses under CO2 and sulfate aerosols are much weaker, only significant in part of Asia and tropical Africa (Fig. 4g and i)."

3. The change in cloud cover is explained by the change in RH. However, there are a lot of other factors affecting clouds (radiation, dynamics, thermodynamics, etc., see Bretherton (2015) and references therein), and I think a more detailed discussion would be helpful. The authors look at vertical velocity and suggest that the change in stability plays less of a role, but it is not clear to me how the conclusion is reached. The estimated inversion strength or lower troposphere stability may be a better predictor for stability.

Response: accepted. We added lower troposphere stability in Fig. 6 and also kept the vertical velocity, as this is reported by some previous studies saying that subsidence could impact cloud cover (e.g., Myers and Norris, 2013). Thus, for the

cloud cover changes, on top of humidity, we also discussed liquid water, moisture flux, dynamics and stability. Some discussions are also included and we also acknowledged that it is impossible to examine all the factors in the current study due to limited output.

We added the following discussion after line 192:
"Another mechanism that has been reported to influence cloud cover is LTS, in which a stable boundary layer could trap more moisture, thereby permitting more low-level clouds (Wood & Bretherton, 2006; Bretherton, 2015). In order to investigate this mechanism, we further analyzed LTS, defined as the difference of potential temperature between 700 hPa and surface (Fig. 6g-i), in which positive anomalies indicate a stronger inversion or weaker lapse rate. The LTS response is again strongest in response to BC forcing (Fig. 6h), with a widespread increase in stability. A previously reported positive correlation between LTS and low-level cloud cover is, nonetheless, only observed in BC source regions (tropical Africa and India) and part of the central US (Fig. 6h). The LTS responses over land are much weaker in response to $CO_2$ and $SO_4$ forcing, with some responses in Africa and India in response to sulfate aerosols (weaker inversion and less cloud). Some other factors have also been suggested to play a role in modifying low-level clouds, such as the diurnal cycle (Caldwell & Bretherton, 2009) and radiative effects of cirrus clouds (Christensen et al., 2013). Due to the limited model output, however, we acknowledge that it is impossible to examine these factors in the current study and it is beyond the scope of our study to probe all possible factors driving the cloud changes. In summary, the above analyses illustrate that the cloud cover changes we see can be primarily explained by RH changes and, to a lesser extent, changes of liquid water content, circulation, dynamics, and stability."

[Figure]

**Figure 2: Same as Fig. 2, but for changes of moisture flux convergence (MFC, a-c), vertical velocity (omega, d-f) and lower tropospheric stability (LTS, g-i) per unit forcing. For vertical velocity (omega), positive anomalies indicate the air is less convective. LTS is calculated as the difference of potential temperature between 700 hPa and the surface. Positive LTS anomalies in g-i indicate stronger inversion or weaker lapse rate.**

4. I have some conservation about including downward LW in the multilinear regression model. It is possible that downward LW change is a result rather than a cause of Tmax change (Tmax change results in changes in boundary layer temperature and moisture, and thus downward LW). In fact, consider the approximation $LW \sim \sigma T^4$, $dLW \sim 4\sigma T^3 dT$, with T=300 K, $dT/dLW \sim 1/(4\sigma T^3) \sim 0.16$, which is very close to the coefficients derived from the regression.

Response: thanks for your demonstration. For the multilinear linear regression, we still prefer to keep the LW component, as dT is directly related with incoming radiation. The aim of the linear regression is to attribute the contribution of those radiation components to Tmax changes and whether the radiative component is forcing or feedback is not important. In fact, the SWCRE we discussed in this study is mainly a feedback process, which is also included in the regression model. Your demonstration here further lends confidence to our regression results.

5. In PDRMIP BC and sulfate are increased by a factor of 10 and 5, respectively. It may be helpful to comment on whether the response is linear for such a large change. The small SWCRE response to sulfate is interesting and somewhat surprising. Given that aerosol direct effect is probably more linear than aerosol

indirect effect, would the authors expect different SWCRE response to historical change in sulfate?

Response: Previous studies show that most aspects of the climate change linearly with climate forcing, including BC. Thus, the large perturbation is unlikely to substantially impact the results and conclusions. We added a sentence in section 2.2 after line 116.

"Previous studies demonstrated that climate changes linearly with climate forcing for various forcing agents, including BC (Hansen et al., 2005; Mahajan et al., 2013)."

For SWCRE response to sulfate change, we do not have a definitive answer to the question of why these are so small or how linear they might be. Based on Fig. 8, the SWCRE change at the surface is not sensitive to cloud cover changes under sulfate aerosol forcing, as both aerosols and clouds scatter solar radiation. Thus, we cannot rule out the possibility that the historical changes are similar to our results.

**Minor:**

1. Please clarify that the paper analyzes SWCRE at the surface in the abstract, the main text, and the figures. It is somewhat confusing because I think SWCRE is more commonly referred to as TOA radiative forcing, and the first paragraph in the introduction describes SWCRE at the TOA.

Response: accepted and clarified where necessary.

2. Eq.(1): What is the time frequency of q and V for calculating the moisture flux?

Response: The time frequency of q and V is monthly, and we added this in the methods section, line 131:

"In Eq (1), q is specific humidity in g kg$^{-1}$, and V is horizontal wind including both zonal and meridional components. All variables have a monthly temporal resolution."

3. Figure 7: Maybe show the fast and slow responses of SWCRE instead of cloud cover, as the paper focuses on SWCRE.

Response: accepted and changed. Here is the new Fig. 7.

[Figure]

**Figure 3: Same as Figure 2 (d-f), but for fast (a-c) and slow responses (d-f) of SWCRE changes per unit forcing.**

---

## Author Comment (AC2) · 16 Jun 2020

**Response to comments #2**

Response: Thanks for your helpful comments. We have made several modifications and implemented the suggestions as described below. We describe a few major changes first, followed by our response to individual comments.

i) Response of cloud liquid water is added.

ii) Response of lower tropospheric stability is added.

iii) SWCRE response for individual models is added to the supporting material.

iv) Replace Fig. 7 with SWCRE response.

This paper is interested in GCM-produced summertime changes in the maximum land temperatures of the NH under perturbed conditions, namely doubled $CO_2$, 10 times more black carbon aerosols and 5 times more sulphate aerosols (subject to model interpretation). Results come from a somewhat outdated database (CMIP5 era models) and the focus is on the SW effects of clouds at the surface (at least initially, later when a prediction model is built LW is added too). I'm not clear what we learn from the analysis. The general consensus since AR5 has been that low clouds provide a positive feedback under $CO_2$ doubling (or quadrupling for that matter), so SWCRE at the surface is expected to be weaker (less cooling at the surface). So, this part is not so new, although I guess one can focus on the effect of this reduced radiative cooling on Tmax. Then there is the aerosol: aerosol changes can change the environment, the circulation, etc, so they can change cloudiness. But they can impact the clouds "faster" through alteration in microphysics (lifetime, optical thickness changes) and this part is not discussed until the conclusions. In any case, the effect of aerosol on (low?) clouds and therefore on land Tmax is not clear-cut since there is also the direct radiative dimming or brightening part that works in conjunction or competition with the cloud effect. So, it's kind of interesting to see results about this, although I imagine people have previously looked at that too. I guess the most intriguing result is that Tmax changes can largely predicted by LOCAL RADIATIVE changes; it was somewhat unexpected to me that this works as well as it does since temperature is also affected by turbulent fluxes and advection (non-local effects). I suggest the authors make a bigger deal of this finding.

Response: We really appreciate the reviewer for speaking so highly for our research. To our best knowledge, CMIP6 models do not have such multi-model inter-comparison project that investigates the climate response to individual forcing

agents yet. Thus, PDRMIP is still the only multi-model project for understanding climate response to individual climate forcings. Our main focus is SW, and LW is included later is because it also impacts surface Tmax. The linear regression aims to quantify the contribution of each radiative component to Tmax. Compared with previous cloud feedback studies, our study has contributions in the following perspectives: 1) better understanding of cloud feedback to individual forcing agents (e.g., stronger response to BC than to GHGs); 2) better understanding on surface SWCRE instead of TOA (e.g., surface SWCRE under sulfate aerosol is much weaker than TOA); 3) quantify their contributions to Tmax, as many previous studies reported this cloud impact on heatwave and drought events, but none of them quantified such impact. We added cloud liquid water analysis in the revised version. For aerosol indirect effect, it is a limitation in PDRMIP study, as the concentrations needs to be fixed. A sentence is added in data section to inform the readers that most of the model have direct effect only in line 104:

"It is noted that only three of the nine models include aerosol-cloud interactions while the remaining ones only have aerosol-radiation interactions. However, this does not impact our main conclusions (see section 4)."

The radiative dimming/brightening effect is already included in $SW_{clear-sky}$ component, whose cooling effect is outweighed by warming effect (Table 2). Fig. 10 and R value indicate that the linear fit works fairly well. However, it is not expected to explain 100% of Tmax changes. Other factors may also play a role, which has been acknowledged in the manuscript, such as line 277:

"It is noted that the radiation change might not explain all Tmax changes, as other factors may come into play. For instance, the temperature response would be different when surface is getting drier under a warmer climate. This is because more net radiation is realized as sensible heat instead of latent heat under drier conditions, which has been suggested to play an important role in recent European heatwaves (Seneviratne et al., 2006; Fischer et al., 2007)."

Here are my main issues with this paper:
1. LW and indirect cloud effects are not discussed until the concluding section.
Response: the main focus of our study is SWCRE. LW impact is small and is not our focus. Thus, we put LW in the discussion section. For aerosol-cloud interactions, it is one of the limitations in the PDRMIP models, so we acknowledge this together with other limitations in the end. However, we added a sentence in the section 2.1

to inform the readers that most of the models only have aerosol-radiation interactions in line 104:

"It is noted that only three of the nine models include aerosol-cloud interactions while the remaining ones only have aerosol-radiation interactions. However, this does not impact our main conclusions (see section 4)."

2. Since the SWCRE effects are mostly attributed to CF changes, the LW downwelling to surface changes should also be broken to clear and LWCRE effects; I mean basically the LW should be treated as the SW and not lumped into a single term in the regression.

Response: the main focus of our study is SWCRE, because SW dominates in the cloud radiative effects. Many published studies (e.g., those in the introduction section) reported that SWCRE plays an important role in amplifying heatwave and drought events world-wide, as cloud cover reduction directly enhance solar heating, thereby raising Tmax. LW effect in these processes is very limited. Our study follows these studies and extends the investigation into the contribution of SWCRE to Tmax. As LWCRE effect is small, we prefer to keep the current regression results.

3. Why are only CF changes considered and not changes in other cloud properties? Optical thickness changes can have impact in SWCRE.

Response: We focus on CF changes because CF could largely explain the SWCRE changes. PDRMIP does not provide output on cloud optical thickness. However, we added cloud liquid water analysis in the revised version (Fig. 4), which is directly related with optical thickness and impact SWCRE (please see section 3.2 in the revised version) in line 173:

"The response of cloud liquid water in the BC experiment could further support this conclusion (Fig. 4h). Liquid water decreases (increases) in regions with decreasing (increasing) cloud cover, following the pattern of RH. As cloud water content directly impacts cloud optical thickness and albedo, such a response may further impact SWCRE (i.e., enhance reflectance in regions showing increasing liquid water and enhance transmittance in regions with decreasing liquid water). However, the liquid water responses under $CO_2$ and sulfate aerosols are much weaker, only significant in part of Asia and tropical Africa (Fig. 4g and i)."

[Figure]

Figure 1: Same as Figure 2, but for relative humidity (a-c), specific humidity (d-f), and cloud liquid water (g-i) at 850 hPa.

4. Only CF changes for low clouds (and the corresponding RH) are considered (if I understand correctly), but for SW cloud at any altitude in the in the atmospheric column can have strong SWCRE effects

Response: We consider low-level clouds because low-level clouds dominate SW changes. We also analyzed the cloud cover changes in 500 hPa and 300 hPa, which show similar changes to those at 850 hPa. We mentioned this in section 3.1 and the figure is included in supporting material (Fig. S5).

[Figure]

Figure S2: same as Fig. 2(d-f) in the main text, but for cloud cover changes at 300 hPa and 500 hPa.

Some minor issues:

1. Clarify from the start that SWCRE refers to surface.
Response: accepted and clarified where necessary.

2. Define changes in SWCRE more formally. For this SWCRE itself has to be defined more formally, i.e., difference between net all-sky and clear-sky fluxes where net = down-up flux. Then you have to take a difference between baseline and perturbed conditions. Just saying that a positive SWCRE change means less cooling is unsatisfying.
Response: accepted and clarified in section 2.2.
"In this study, we focus on the SWCRE at the surface in the low and mid-latitudes during boreal summer months (June-July-August, JJA hereafter), which is calculated as the difference in the SW radiative flux at the surface between all-sky and clear-sky conditions (Ramanathan et al., 1989)……. Changes in SWCRE are obtained by subtracting the control simulations from the perturbations using the data of the last 20 years in each coupled simulation."

3. I find the discussion between fast and slow feedbacks a bit superficial. Land responds to fast feedbacks, but for slow feedbacks the SST responds as well and that's what will drive circulation changes. For slow feedbacks it makes more sense to look at TOA quantities. When it comes to direct radiative effect of aerosol, TOA and SFC changes are distinct for absorbing (BC) vs non-absorbing aerosols (sulphate).
Response: we agree that the slow response is controlled by global mean temperature change (including SST). However, the aim of this part is to give a qualitative picture that the cloud response is mainly due to fast response, slow response or both. What specific process that drives these slow responses is not our focus. Following another reviewer's comment, we replaced the Fig. 7 of fast and slow cloud cover response with SWCRE changes at the surface.

[Figure]

Figure 3: Same as Figure 2 (d-f), but for fast (a-c) and slow responses (d-f) of SWCRE changes per unit forcing.

4. Why not use the same colorbar in Fig. 2 for normalized forcing change and cloud fraction change to make comparison easier (of course range of values can be different)?

Response: changed to same colorbar.

5. By showing only MMM results and nothing about model spread we have no idea how much the models diverge in predictions. Not sure there is an easy way to convey that.

Response: As these are spatial maps, we could not figure out a way of showing inter-model spread at this moment. So we just follow the traditional way by showing MMM results. In fact, the uncertainty bars in the bar plot (Fig. 3) could shed some light on the inter-model spread of the results. For CO2 and BC, the results are quite consistent across the models and for SO4, even the sign of change is uncertain and thus, a larger range is seen. These results are further illustrated by the individual model response, which has been included in the supporting material (Fig. S2-S4).

[Figure]

**Figure S4: SWCRE changes per unit forcing by individual models for the CO₂ experiment.**

[Figure]

**Figure S5: SWCRE changes per unit forcing by individual models for the BC experiment.**

[Figure]

**Figure S6: SWCRE changes per negative forcing for the sulfate aerosol experiment.**

6. I imagine the radiative treatment of aerosol differs widely among models. Not discussed. When you change emissions instead of concentrations directly, divergence is introduced too.

Response: the readers could refer to the literature documenting each model in Table 1 for detailed radiative treatment of aerosols, as it is nearly impossible to discuss them one by one. We added the SWCRE changes for individual models into supporting material (Fig. S2-S4; see the response above). The main features are consistent across models and not sensitive to model setup (e.g., emission, concentration or radiative treatment), indicating that our results are fairly robust. We added these in section 3.1 line 144:

"When it comes to individual model response (Fig. S2-S3), these patterns are also consistent across at least eight of the nine models and are not very sensitive to the model setup (emission-based or concentration-based)."

7. I also imagine that the base state of the models is quite different too. Care to comment?

Response: The multi-model mean value of SWCRE in the base run is -57.9±1.8 W m$^{-2}$ (MMM±1 standard error). The spatial patterns are fairly consistent across the models, with strong SWCRE in tropical regions and mid-to-high latitudes and weaker SWCRE in subtropics, regions generally with less clouds. We added this

figure in the supporting material (Fig. S1) and also mentioned this in section 2.2 in line 111:

"The base state of SWCRE in each model is shown in Fig. S1, with a multi-model mean (MMM) value of -57.9±1.8 W m$^{-2}$ (MMM±1 standard error). The spatial patterns are fairly consistent across the models, with strong SWCRE in tropical regions and mid-to-high latitudes and weaker SWCRE in subtropics, regions generally with less clouds."

[Figure]

**Figure S7: SWCRE in the base climate for each model. The global mean values are shown in the upper-right corner.**